# Exploring novel immunotherapy biomarker candidates induced by cancer deformation

**Se Min Kim**[1], **Namu Park**[2], **Hye Bin Park**[3], **JuKyung Lee**[3], **Changho Chun**[4,5], **Kyung Hoon Kim**[4], **Jong Seob Choi**[6], **Hyung Jin Kim**[7], **Sekyu Choi**[8], **Jung Hyun Lee**[9,10]*

1 Life Science and Biotechnology Department (LSBT), Underwood Division (UD), Underwood International College, Yonsei University, Sinchon, Seoul, Korea, 2 Department of Biomedical Informatics & Medical Education, University of Washington, Seattle, Washington, United States of America, 3 Digital Health Care Research Center, Gumi Electronics and Information Technology Research Institute (GERI), Gumidaero, Gumi, Gyeongbuk, South Korea, 4 Department of Bioengineering, University of Washington, Seattle, Washington, United States of America, 5 Department of Rehabilitation Medicine, University of Washington, Seattle, Washington, United States of America, 6 Division of Advanced Materials Engineering, Kongju National University, Chungnam, South Korea, 7 School of Electrical & Electronic Engineerin, Ulsan College, Ulsan, Korea, 8 Department of Life Sciences, Pohang University of Science and Technology (POSTECH), Pohang, Republic of Korea, 9 Department of Dermatology, School of Medicine, University of Washington, Seattle, Washington, United States of America, 10 Institute for Stem Cell and Regenerative Medicine, University of Washington, Seattle, Washington, United States of America

* jlee24@uw.edu, hyunlee228@gmail.com

**Data Availability Statement:** All relevant data are within the paper.

**Funding:** In our Financial Disclosure, we state that this research received partial support from The Elsa U Pardee Foundation (to J.H.L) and the MCC

## Abstract

Triple-negative breast cancer (TNBC) demands urgent attention for the development of effective treatment strategies due to its aggressiveness and limited therapeutic options [1]. This research is primarily focused on identifying new biomarkers vital for immunotherapy, with the aim of developing tailored treatments specifically for TNBC, such as those targeting the PD-1/PD-L1 pathway. To achieve this, the study places a strong emphasis on investigating Ig genes, a characteristic of immune checkpoint inhibitors, particularly genes expressing Ig-like domains with altered expression levels induced by "cancer deformation," a condition associated with cancer malignancy. Human cells can express approximately 800 Ig family genes, yet only a few Ig genes, including PD-1 and PD-L1, have been developed into immunotherapy drugs thus far. Therefore, we investigated the Ig genes that were either upregulated or downregulated by the artificial metastatic environment in TNBC cell line. As a result, we confirmed the upregulation of approximately 13 Ig genes and validated them using qPCR. In summary, our study proposes an approach for identifying new biomarkers applicable to future immunotherapies aimed at addressing challenging cases of TNBC where conventional treatments fall short.

## 1. Introduction

Cancer immunotherapy has a rich history, dating back to observations by physicians like Fehleisen and Busch, who witnessed tumor regression following erysipelas infections [2]. Over time, pioneers such as William Bradley Coley laid the foundation for this field, leading to fundamental immunology discoveries in 1967 that unveiled the crucial role of T cells in immune

Patient Gift Fund (to J.H.L). The funders had no role in the study design, data collection and analysis, decision to publish, or preparation of the manuscript.

**Competing interests:** The authors have declared that no competing interests exist.

responses [3]. The 2018 Nobel Prize awarded to James Allison and Tasuku Honjo for their work on checkpoint molecules has catalyzed the development of promising treatments, including checkpoint inhibitors, CAR T-cells, and oncolytic viruses, reshaping the future of cancer therapy [4].

Immune checkpoint inhibitors (ICIs) have transformed cancer treatment, instilling renewed hope in patients facing various malignancies [5]. The varied and occasionally unpredictable responses of patients to these therapies emphasize the pressing necessity for predictive biomarkers that can identify those with the highest likelihood of benefiting [6]. Despite the advantages of ICI-based therapies, the majority of patients experience disease progression after initially promising responses [7]. The current optimal strategy for enhancing the effectiveness of ICIs and gaining a deeper understanding of the mechanisms driving cancer resistance is to explore novel, effective, and well-tolerated combination therapies [8]. Nonetheless, while discovering new ICIs remains crucial, only a few candidates have been identified thus far [9].

Among immunotherapies, antibody-based treatments have a rich history dating back to the late 19th century when pioneering scientists like Paul Ehrlich, Emil von Behring, and Kitasato Shibasaburo made groundbreaking discoveries regarding antibodies [10]. These small proteins have since become indispensable in treating various diseases, including cancer. Monoclonal antibodies, a significant advancement made possible by the pioneering work of Milstein and Köhler in the 1970s, have revolutionized this field [11]. They are engineered to specifically target antigens and can be produced at a large scale, playing a pivotal role in modern immunotherapy [12].

Monoclonal antibodies in cancer therapy can either tag cancer cells for immune system destruction or disrupt signaling pathways that drive cancer cell growth [13]. Notably, rituximab stands as a milestone, being the first FDA-approved antibody for cancer therapy in 1997, marking a significant advancement in cancer immunotherapy and providing new hope for patients battling this devastating disease [14].

Furthermore, antibodies like trastuzumab have been customized to target specific types of cancer cells, effectively inhibiting their growth [15]. Some antibodies are also combined with other agents to enable precise targeting and destruction of cancer cells [16]. An up-and-coming area of antibody-based cancer research revolves around checkpoint inhibitors, with several FDA-approved drugs [5]. James Allison's groundbreaking work in the 1980s paved the way for immune checkpoint blockade therapies [17]. Notably, the development of the first CTLA-4 blocking antibody in 1996 and the subsequent FDA approval of ipilimumab in 2011 marked a significant breakthrough, showing remarkable success in treating melanoma and other types of cancer [18].

Subsequent approvals of PD-1 and PD-L1 inhibitors, such as nivolumab, pembrolizumab, atezolizumab, durvalumab, and avelumab, have significantly improved outcomes across various cancers [19]. These inhibitors target immune checkpoint molecules expressed on tumor-infiltrating lymphocytes and specific tumor cells, enabling the immune system to mount a more effective defense against cancer [20–22]. As a result, survival rates have significantly improved, especially in melanoma and lung cancer, providing newfound hope for many patients [23]. Overall, antibody-based therapies have transformed the landscape of cancer treatment, demonstrating their potential to save lives and redefine the approach to cancer care [24].

The most significant advancement in cancer treatment over the past decade has undoubtedly been the introduction of T cell-targeted immunomodulators that block immune checkpoints like CTLA-4 and PD-1 or PD-L1 [25–27]. In 2011, ipilimumab became the first authorized antibody to block an immune checkpoint, specifically CTLA-4 [28]. Following the introduction of CTLA-4 blocking antibodies, single-cloned antibodies targeting anti-PD-1

(pembrolizumab and nivolumab) and anti-PD-L1 (atezolizumab and durvalumab) were developed promptly [29–32]. Anti-PD-1/PD-L1 antibodies have emerged as some of the most widely prescribed anticancer therapies. T-cell-targeted immunomodulators are now used as standalone treatments or in combination with chemotherapy as first or second-line options for approximately 50 different cancer types [7,33]. Currently, there are more than 3000 active clinical trials assessing T cell modulators, accounting for approximately two-thirds of all oncology trials [34,35].

Invasive breast cancer is classified into five primary molecular subtypes based on gene expression patterns affecting cell behavior. These subtypes include Luminal A (ER+/PR +/HER2-), associated with a favorable prognosis; Luminal B (ER+/HER2- with high Ki-67 or PR-); Luminal B-like (ER+/HER2+ with variable Ki-67 and PR status), showing a slightly worse prognosis; HER2-enriched (ER-/PR-/HER2+), often effectively treated with targeted therapy; and Triple-negative or basal-like (ER-/PR-/HER2-), recognized as more aggressive and prevalent in specific groups like BRCA1 mutation carriers [36]. These diverse molecular subtypes underscore the critical need to discover new immunotherapy biomarkers for optimizing breast cancer treatment decisions.

In conclusion, despite challenges in predicting the efficacy of ICI treatments, ongoing research endeavors are illuminating numerous potential biomarkers and strategies that can enhance our capacity for personalized immunotherapy [37–39]. These advancements in biomarker discovery and validation play a crucial role in the future of precision immuno-oncology to improve outcomes for cancer patients undergoing ICI therapies [40–42]. To fully leverage the advantages of precision immuno-oncology, it is essential that not only the discovery of new ICIs but also comprehensive research into the molecular structures of ICIs and their mechanisms of intracellular interactions, especially their specific interactions with immune cells, are conducted simultaneously. Predictive cancer immunotherapy biomarkers have the potential to facilitate personalized treatment strategies, ultimately affording cancer patients a more optimistic prognosis by providing enhanced outcomes, tailored therapies, and the promise of a brighter future.

## 2. Materials and methods

### RNA sequencing analysis

In this current study, we utilized the Total RNA sequencing service provided by BGI (https://www.bgi.com/us/home), which had been previously employed in our previous research [43]. We identified differentially expressed genes among cells migrated through the microchannel using DESeq2 (ver 1.32.0). The bulk RNA-Sequencing dataset included three sequencing results from TW1 cells and three sequencing results from the original triple-negative cells as a control (TW1-1, TW1-2, TW1-3, Ori-1, Ori-2, Ori-3). Genes with significantly increased/decreased expressions were selected with adjusted p-values < 0.05 and were grouped into upregulated genes (log fold change > 1) and downregulated genes (log fold change < -1). Since our goal was to identify a new set of candidate genes as immune checkpoint regulators, we further filtered the differentially expressed genes based on their cellular localization. We obtained localization information from www.genecards.org in the 'Localization' section for each gene, and if the confidence score were > = 4 in the 'plasma membrane' or 'extracellular' compartments, the genes were selected. Next, within the filtered group of genes, we focused on genes that had immunoglobulin (including immunoglobulin-like) domains, according to https://www.ebi.ac.uk/interpro/.

## Cell culture

In previous research [43], TW-1 and TW-2 cell lines were established from MDA-MB-23 cells and cultured using DMEM supplemented with 10% FBS (Gibco, 10270) and 1% penicillin-streptomycin (Gibco, 15140122) in a 5% CO2 incubator at 37˚C.

*Quantitative real-time PCR*: We extracted total RNA from cells using Trizol (Invitrogen). After checking RNA purity and concentration, RNA was reverse transcribed to cDNA using the iScript SuperMix reagent (BioRad). Primers (IDT) were diluted in nuclease-free water with PowerUp SYBR Green master mix (Applied Biosystems) and qPCR performed on the Applied Biosystems 7300 machine. The following primers were used human β-actin: forward, 5' -ACTCTTCCAGCCTTCCTTCC- 3', reverse, 5' -CAATGCCAGGGTACATGGTG- 3'; human CRLF2: forward, 5'– GGTGATGTGGTCACAATCGG—3', reverse, 5'– TGACAGTGGTGTGTCCATCA—3'; human SORCS2: forward, 5'– CCTGTGCGACTACGGATTTG—3', reverse, 5'– CACACGTTGGACACCACTTT—3'; human SORL1: forward, 5'– ACCTGTCTTCGCAACCAGTA—3', reverse, 5'– GGTGGTAGGGCAGTTTCTCT—3'; human IL7R: forward, 5'—GTCTATCGGGAAGGAGCCAA—3', reverse, 5'– TCATACATTGCTGCCGGTTG—3'; human CSF2RA: forward, 5'—TGAGTTACCACACCCAGCAT—3', reverse, 5'– TTACTGAGCCTGGGTTCCAC—3'; human LRIG1: forward, 5'– TTCACATAAGGCCAGGCTCA—3', reverse, 5'– CATCCTTCTGCCAGGCAATC—3'; human SIRPA: forward, 5'– CCCTCTACCTCGTCCGAATC—3', reverse, 5'- GGGCTCATGCAACCTTGTAG—3'; human MDGA1: forward, 5'-AGCCACCCACATAATCACCA—3', reverse, 5'—GGCTACTCTAGGGCAGGAAG—3'; human IL1R1: forward, 5'- CTTGCCTTTCCACCTGCTTT—3', reverse, 5'—CTAGGCCCTGTCTGTTGGAA—3'; human PSG5: forward, 5'- CTAACCCACCGGCAGAGTAT—3', reverse, 5'—AGGAGCAGAGACTTCGACTG—3'; human PDGFRB: forward, 5'-GCCAGAGCTTGTCCTCAATG—3', reverse, 5'—TGTCTAGCCCAGTGAGGTTG—3'; human CD33: forward, 5'- AGGAGATGGCTCAGGGAAAC—3', reverse, 5'—TAGGGTGGGTGTCATTCCTG—3'; human LRRN1: forward, 5'- AGGTAGAAGGCCAGTTCCAC—3', reverse, 5'—TGGCACCTGTCTGATGGAAT—3'; human LILRB1: forward, 5'-ATGCGTCTCTGCTGATCTGA—3', reverse, 5'—CCTGCTCTGTGGATGGATGA—3'; human IL13RA2: forward, 5'- TTAAACCTTTGCCGCCAGTC—3', reverse, 5'—CAACTGTAGCAGTCACCAAGG—3'; human NDNF: forward, 5'-TGACAAGCTCCGTACCTGTT—3', reverse, 5'—TGTTCCTTTCCTGAGTGCCT—3'. Relative RNA levels were calculated from the Ct values.

## 3. Results

### Variability in genetic changes observed in cancer deformations

TNBC, constituting 15–20% of breast carcinomas, is an aggressive subtype with a notably unfavorable prognosis in comparison to other subtypes. Standard treatment includes chemotherapy, but only 30% show a complete response and many face recurrence and mortality even after surgery and radiotherapy. Recent research has uncovered the potential for immune responses in TNBC, characterized by higher tumor mutational burden (TMB), tumor-infiltrating lymphocytes (TILs), and PD-L1 expression. Currently, PD-1/PD-L1 and CTLA-4 inhibitors are approved for clinical use. However, optimizing immunotherapy, including timing, combinations, and predictive biomarkers, requires further research to improve TNBC patients' outcomes significantly [39]. Therefore, a strategy has been devised to identify novel immunotherapy biomarkers specific to malignant breast cancer in order to address the issues of resistance and non-responsiveness to PD-1/PD-L1 blockade therapy. In alignment with strategies

employed across diverse cancer types, breast cancer therapeutics has primarily leaned on deploying PD-1 or PD-L1 blocking antibodies as a central immunotherapeutic approach [44–46]. Immune checkpoints, exemplified by the PD-1/PD-L1 axis, are strategically positioned within the tumor microenvironment and lymphoid organs, providing cancer cells the means to co-opt these checkpoints and dampen anti-tumor immune responses [47,48]. Our research aims to predict novel biomarkers by analyzing genes containing immunoglobulin-like domains (Ig domains), akin to PD-1 or PD-L1, whose expression patterns undergo alterations due to cancer deformation. Genes harboring Ig domains are pivotal in immune checkpoint inhibitor (ICI) therapy. By investigating the expression levels of genes containing Ig domains that are modified due to cancer transformation, this study aims to enhance our comprehension of the immune evasion mechanisms in malignant breast cancer. Fig 1A illustrates our strategy for developing more effective immunotherapy for Triple Negative Breast Cancer (TNBC).

Our previous study designated MDA MB 231 Original cells as 'Ori' and cells that underwent deformation after passing through a Transwell with a 3μm pore size as 'TW1.' Subsequently, cells that underwent a second deformation by passing through a Transwell were labeled as 'TW2.' Since TW1 and TW2 exhibited similar malignant characteristics, we conducted RNA sequencing exclusively on Ori and TW1 cells. Using a heatmap, we then examined the overall gene expression changes across all genes (Fig 1B) [43]. Using RNA sequencing data, we conducted a comparative gene expression analysis between Ori and TW1 cells. Fig 2B illustrates discernible alterations in gene expression. However, scrutinizing the top 5 genes displaying the most significant changes presented a challenge to consistently discern noteworthy signaling pathways or pathways with shared functional implications (Fig 1C and 1D). Consequently, our research shifted its focus towards genes harboring immunoglobulin (Ig) domains that are located in the cell plasma membrane and exhibit extracellular expression.

## Expression patterns of ig genes in cancer deformations

To investigate and identify novel biomarkers associated with emerging immunotherapy strategies, we have adopted an innovative approach. In the context of immunotherapy, our decision was to concentrate on genes responsible for encoding proteins featuring immunoglobulin (Ig) domains, much like those expressed on the cell surface, such as PD-1 or PD-L1. Instead of delving into genes linked to intracellular signaling or metabolic pathways, we specifically scrutinized genes falling within this category that displayed distinctive expression patterns, as exemplified in Fig 2A. This choice was grounded in the fact that Ig-like domains exhibit a substantial degree of similarity, heightening the potential for the discovery of proteins like PD-L1, which can hinder the functioning of T cells or immune cells. Conversely, signaling cascades and metabolic processes can undergo subtle variations contingent on cell types or environmental factors, prompting us to forego further consideration of these aspects in this study.

While the human leukocyte antigen (HLA) system exhibits substantial polymorphism, aberrations in HLA expression can profoundly impact antigen processing and immune recognition, potentially thwarting tumor immune evasion [49]. Therefore, HLA molecules are a crucial marker in cancer immunotherapy due to their vital role in efficiently binding to antigen epitopes [50]. This has led to increased interest in the expression levels of HLA molecules, as they play a central role in cancer immunotherapy [51]. The primary role of HLA molecules in inducing and regulating immune responses, along with selecting T-cell repertoires, makes increasing the expression of HLA molecules for cancer cell elimination highly efficient in enhancing T-cell attack against cancer cells. Consequently, HLA genes are critically important in cancer cell immune evasion. Therefore, the downregulation of HLA genes due to cancer

**A**

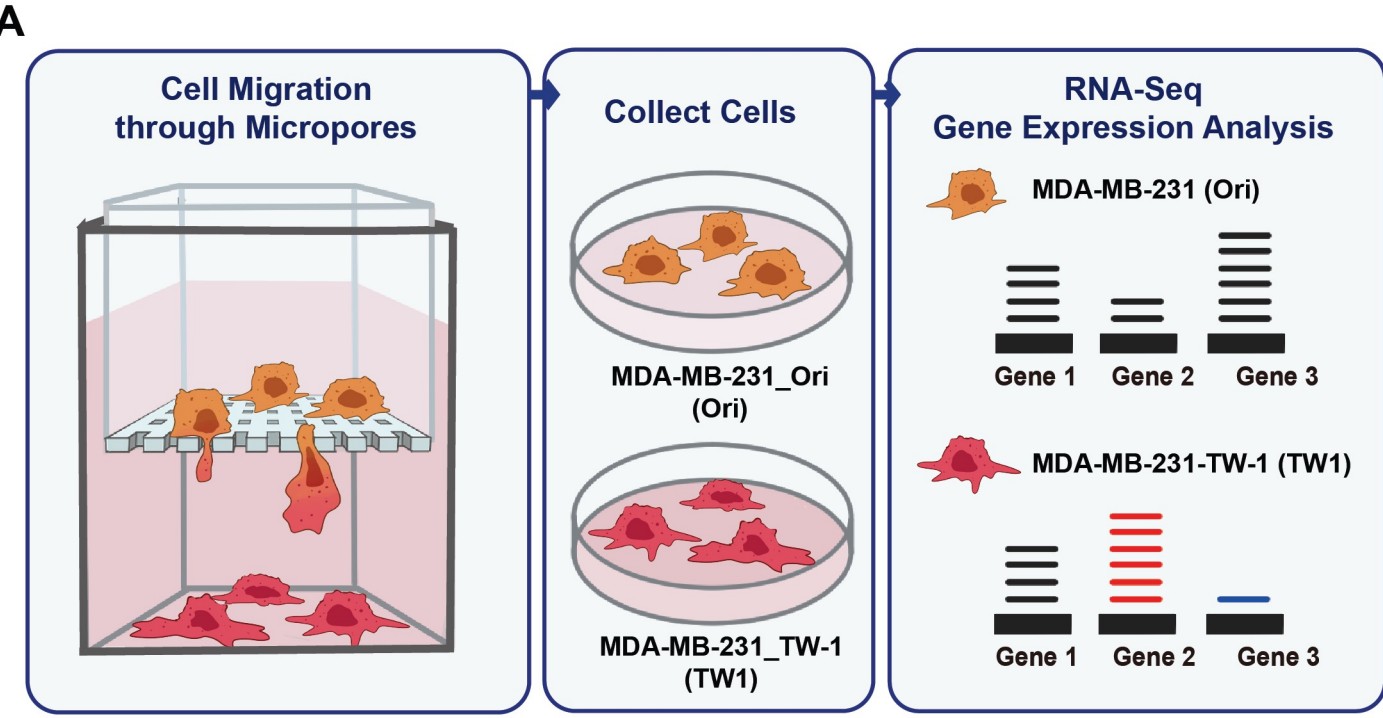

**B**

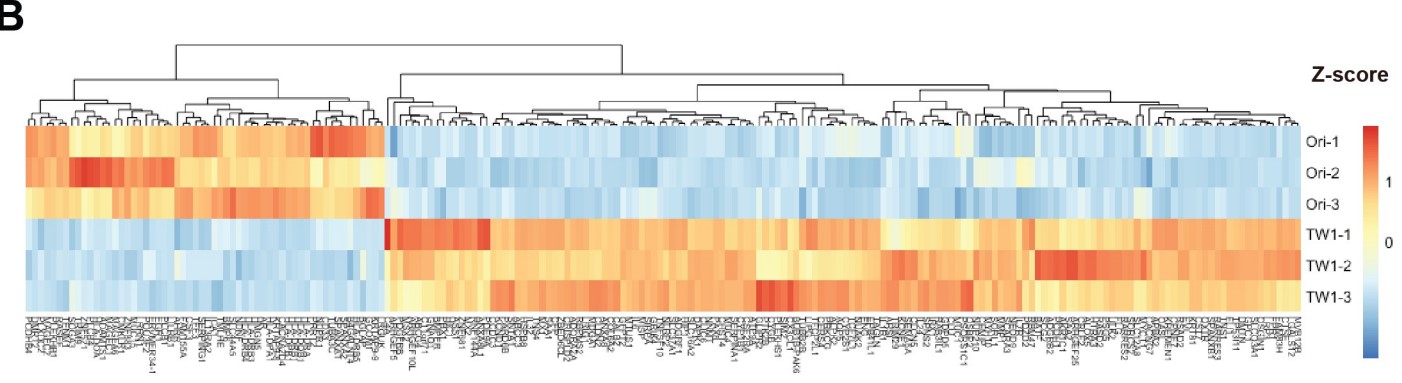

**C**

**The top 5 genes among the upregulated 148 genes**

| # | Gene | baseMean | log2FC | lfcSE | P-value | Adjusted P-value |
|---|------|----------|--------|-------|---------|------------------|
| 1 | TP53I11 | 63.373 | 6.603 | 0.756 | 2.48E-18 | 1.33E-15 |
| 2 | SEMA5A | 5.296 | 5.883 | 1.381 | 2.03E-05 | 9.33E-04 |
| 3 | CX3CL1 | 9.634 | 5.759 | 1.302 | 9.66E-06 | 4.88E-04 |
| 4 | RNF144A | 9.035 | 5.666 | 1.296 | 1.22E-05 | 0.000601 |
| 5 | LGALS12 | 4.246 | 5.566 | 1.419 | 8.81E-05 | 0.003004 |

**D**

**The top 5 genes among the downregulated 58 genes**

| # | Gene | baseMean | log2FC | lfcSE | P-value | Adjusted P-value |
|---|------|----------|--------|-------|---------|------------------|
| 1 | GCOM1 | 6.991 | -6.205 | 1.37 | 5.90E-06 | 3.27E-04 |
| 2 | KRTAP9-8 | 5.995 | -5.991 | 1.351 | 9.30E-06 | 0.000474 |
| 3 | LILRB1 | 3.036 | -5.013 | 1.504 | 8.56E-04 | 0.018017 |
| 4 | KRTAP9-9 | 2.477 | -4.709 | 1.582 | 2.91E-03 | 4.40E-02 |
| 5 | DMRT2 | 36.170 | -4.667 | 0.589 | 2.31E-15 | 9.36E-13 |

**Fig 1.** A, Schematic Illustration Showing the Sample Preparation Process B, Heatmap illustrating variations in differentially expressed genes C, The top 5 genes among the 148 upregulated genes. D, The top 5 genes among the 148 downregulated genes.

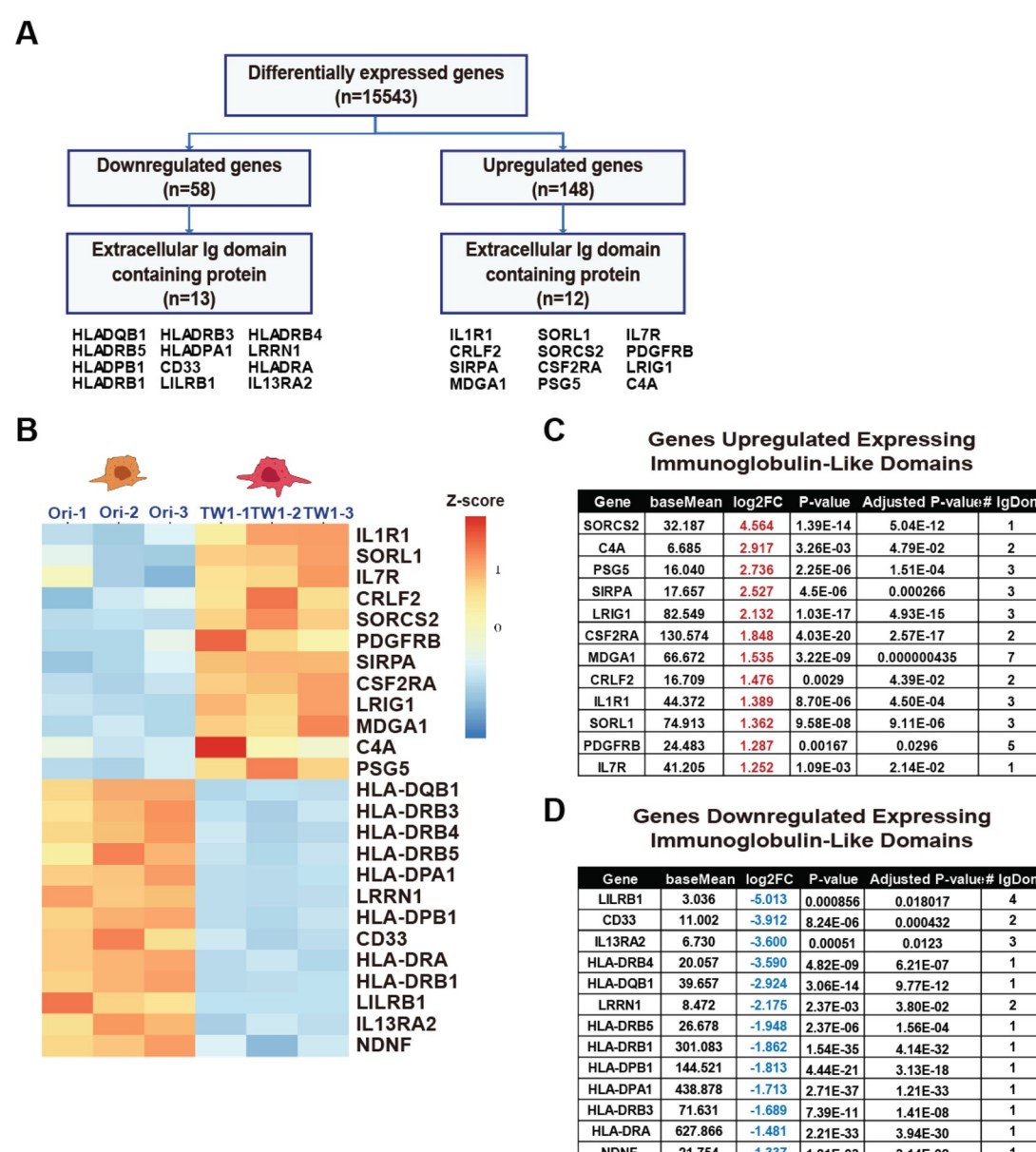

**Fig 2.** A, Schematic diagram for distinguishing the extracellular Ig domain. B, RNA Sequencing-Derived Heatmap of Genes with Distinct Ig-Like Domains. C, List of genes with increased expression having Ig-like domains. D, List of genes with decreased expression having Ig-like domains.

deformation in our results signifies a predominant occurrence of immune evasion by cancer cells. This suggests that our experimental approach is well-suited for discovering biomarkers for novel immunotherapy against malignant cancers, as illustrated in Fig 2B. In Fig 2B, the TW1 cells that experienced deformation displayed downregulation of several genes, including HLA genes. Among the most significantly downregulated genes were CD33, LILRB1, LRRN1, IL13RA2, and NDNF, with CD33, IL13RA2, and LILRB1 being well-known genes associated with immune activity. The prominence of these genes in the context of immune evasion by cancer cells highlights the need for a more comprehensive investigation of the upregulated genes.

To delve deeper into our investigation of upregulated genes containing Ig domains as potential novel immunotherapy biomarkers in metastatic breast cancer cells, our analysis revealed a diverse array of genes, including SORCS2, C4A, PSG5, SIRPA, LRIG1, CSF2RA, MDGA1, CRLF2, IL1R1, SORL1, PDGFRB, and IL7R. Within this set of genes, a subset, notably CSF2RA, CRLF2, IL1R1, IL7R, and PDGFRB, exhibited intricate connections with the JAK-STAT pathway, which is crucial for the viability of cancer cells. This intriguing association, reminiscent of the role of PD-L1 in immune evasion by cancer cells, underscores the complexities of cancer-immune interactions and suggests a potential avenue for targeted therapeutic interventions, as depicted in Fig 2B. We have compiled a table presenting data from the RNA sequencing results for upregulated and downregulated genes containing Ig domains. Our primary emphasis is on the top of genes within each category, and the table also includes information about the number of Ig domains associated with each gene (Fig 2C and 2D).

## Ig genes upregulated due to cancer deformation

To explore the upregulated genes, we identified 11 genes with upregulated Ig domains through RNA sequencing analysis. These 11 genes exhibited a consistent trend of increased expression in TW1 compared to Ori, as illustrated in the violin plot generated from the RNA sequencing results (Fig 3A). There is a potential for these genes to be considered as targets for immunotherapy, as they possess one or more Ig domains. Interestingly, the subsequent qPCR validation yielded clear and consistent results, confirming a significant increase in the expression of all these genes. The upregulation of these genes, consistent with the RNA sequencing results, suggests their interaction with external immune cells due to their expression on the cell membrane. Among the 11 genes listed here, PSG5 is the only gene that possesses Ig domains and is also secreted outside the cell. However, we included it in the list since even when secreted, it could disrupt the activity of immune cells. While RNA sequencing was performed exclusively on Ori and TW1 samples, qPCR validation included TW2 samples as well. Due to further cell deformation, TW2 cells likely represent a subline with a more malignant phenotype. Through qPCR validation experiments, we confirmed the expression patterns of the 11 upregulated genes in Ori, TW1, and TW2 (Fig 3B). Our approach allows us to propose the potential for new immunotherapy biomarkers more concretely by validating the expression patterns of genes containing Ig domains that undergo dramatic changes. Interestingly, through qPCR validation, we were able to confirm that CRLF2, SORCS2, SORL1, IL7R, CSF2RA, LRIG1, SIRPA, MDGA1, IL1R1, PSG5, and PDGFRB all exhibited an increasing trend in expression during cancer deformation (Fig 3B). This indicates the high accuracy of our RNA sequencing results in depicting gene expression patterns. Furthermore, this suggests the potential for representing the changes in cells influenced by cancer deformation in a highly stable manner. This finding underscores the need to assess the suitability of a uniform treatment approach for both primary and metastatic cancers in the context of immunotherapy. It highlights the significance of systematically evaluating tailored treatment strategies throughout different stages of cancer progression.

## Ig genes downregulated due to cancer deformation

To further validate our findings regarding new Ig genes, we investigated the presence of Ig genes beyond those within the HLA family. The RNA sequencing analysis confirmed that additional genes, specifically CD33, LILRB1, LRRN1, IL13RA2, and NDNF, exhibited a decreasing trend as Ig genes, as illustrated in the violin plot in Fig 4B. Subsequently, in the following experiments, we confirmed whether these five genes that appeared to be downregulated exhibited a decreasing expression pattern in Ori, TW1, and TW2 through qPCR. Surprisingly,

## A  Violin Plot of Upregulated Genes

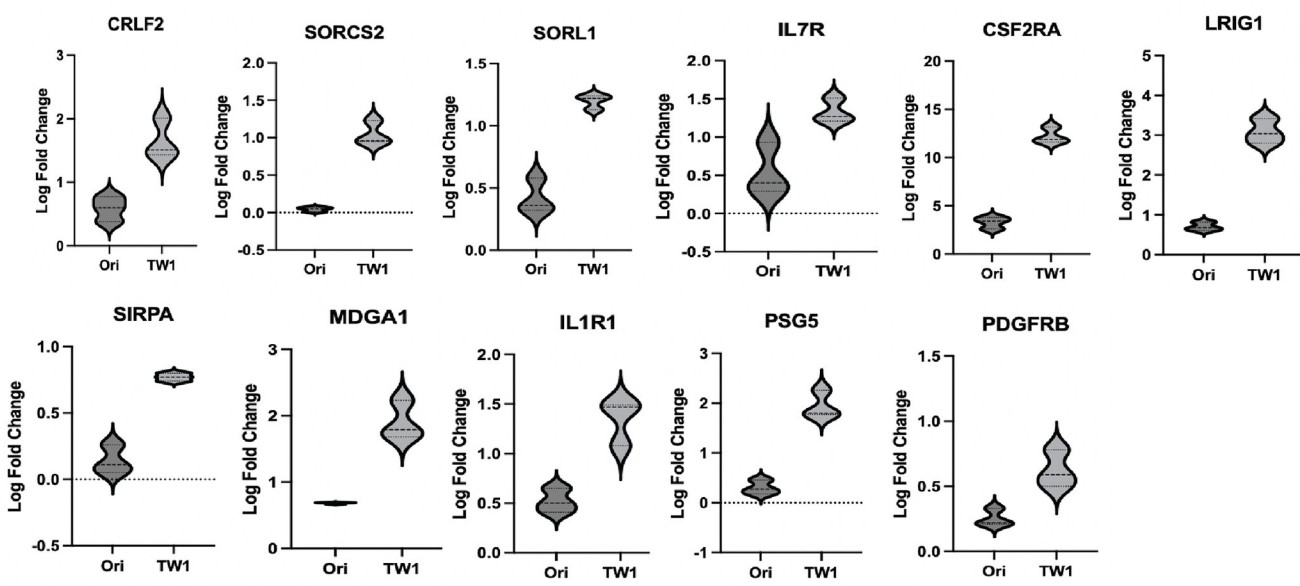

## B  Validation via qRT-PCR

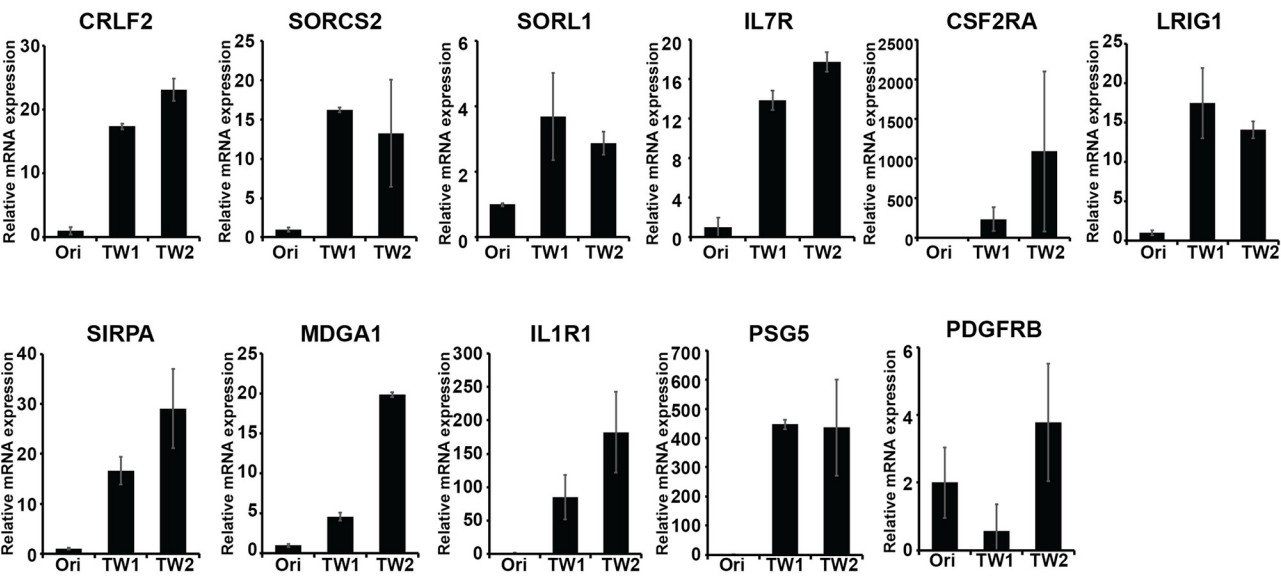

**Fig 3.** A, Violin plot constructed based on RNA sequencing results for genes with increased expression containing Ig-like domains. B, Based on the results of the Violin plot, gene expression changes were validated through qRT-PCR in Ori, TW1, and TW2 samples.

unlike the consistent upward trend observed in all upregulated Ig genes identified through RNA sequencing, downregulated Ig genes, excluding CD33 and LILRB1, displayed an increasing trend (Fig 4B). This highlights the necessity for additional validation methods, such as qPCR, for the downregulated Ig genes identified in RNA sequencing. This discrepancy challenges the reliability of solely relying on sequencing data for biomarker identification.

## A Violin Plot of Downregulated Genes_MHC

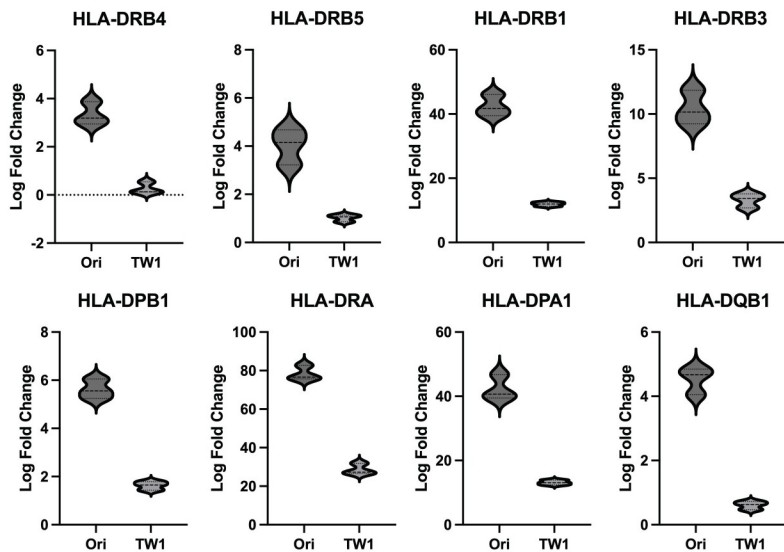

## B Violin Plot of Downregulated Genes

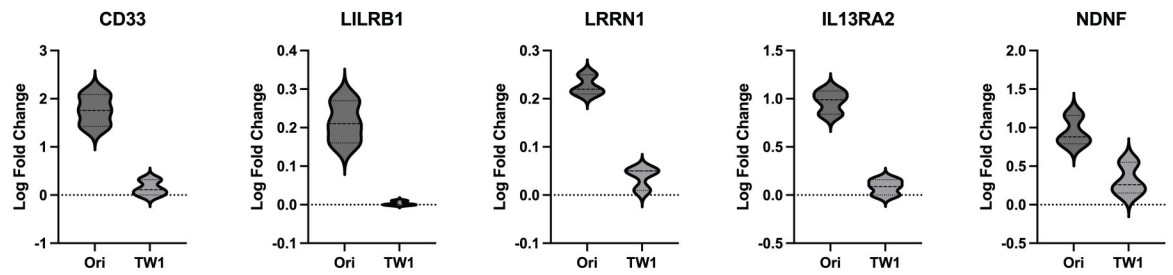

## C Validation via qRT-PCR

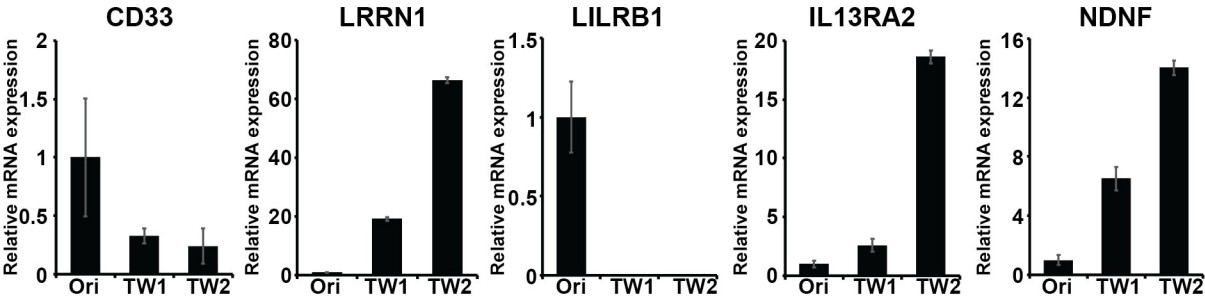

**Fig 4.** A, Violin plot generated using RNA sequencing data to visualize genes with reduced expression that contain Ig-like domains. B, Based on the results from the Violin plot, the reduction in gene expression was confirmed through qRT-PCR in the Ori, TW1, and TW2 samples.

Consequently, we have identified Ig genes that increase in response to cancer deformation among those initially identified as downregulated in RNA sequencing.

We used a schematic illustration to visually represent the characteristics of the extracellular domains of genes that consistently showed increased expression levels in both RNA sequencing and qPCR experiments (Fig 5A). We have compiled a schematic illustration summarizing the Ig genes that have undergone changes in expression levels due to TNBC deformation (Fig 5B). The accumulation of this information will contribute to discovering more immunotherapy treatment targets and enhance the efficacy of treatments.

## 4. Discussion

Treatment strategies for TNBC have traditionally relied heavily on chemotherapy due to the absence of hormone receptors and HER-2 amplification, which limit the availability of targeted therapies and contribute to limited treatment success [52]. The emergence of immunotherapy, particularly immune checkpoint inhibitors (ICIs), has provided a glimmer of hope in the TNBC landscape [53]. However, it's important to note that the efficacy of ICIs as monotherapy has been somewhat modest, benefiting only a subset of TNBC patients [54]. To fully unlock the potential of immunotherapy and make significant advances in TNBC treatment, the development of novel immune combination therapies is crucial. These combination strategies hold great promise for enhancing treatment efficacy and expanding the group of patients who can benefit.

In this research, our objective is to uncover the unexpected strategies employed by cancer cells to evade the immune response induced under the artificial conditions of what we refer to as "cancer deformation." Our initial focus was on Ig genes like PD-L1. Immunoglobulins are highly conserved, so even a portion of their function could have common functionalities. We validated all Ig genes altered by cancer deformation in MDA-MB-231 cells through bulk RNA sequencing. While our results did not unveil well-known immune checkpoint inhibitors, they did expand the possibilities for discovering immunotherapy treatment targets, driven by the increased presence of approximately 13 Ig genes. Further research will elucidate whether these genes interact with immune cells or contribute to the aggressive growth of cancer cells themselves. Furthermore, investigating the mechanisms behind the decrease in the essential HLA family during the cancer deformation process, which is crucial for immune evasion, is expected to be an intriguing avenue for future research. Our study demonstrates the potential utility of our "cancer deformation" method for analyzing cancer-specific or common Ig genes in various cancer types, and it is likely to find continued application in such investigations.

## 5. Conclusions

The immunoglobulin (Ig)-like domain is a common protein feature resembling the structure of immunoglobulins, typically composed of 70–110 amino acids with a central disulfide bond [55]. There are various subtypes of Ig-like domains, including immunoglobulin constant (C-) and variable (V-) types, found in a wide range of proteins like immunoglobulins, T-cell receptors, major histocompatibility complex (MHC) proteins, and cell-surface receptors[56]. PD-L1, also known as B7-H1 or CD274, is a ligand for the PD-1 receptor. Initially identified as B7-H1, it was found to have inhibitory effects on T cells by inducing IL-10 [57]. Structurally, PD-L1 is a 290-amino acid protein encoded by the Cd274 gene, featuring two extracellular IgV- and IgC-like domains, a transmembrane domain, and a short intracellular domain [58]. PD-L1 is constitutively expressed on various cell types, including antigen-presenting cells, T cells, B cells, monocytes, and epithelial cells, with potential upregulation in response to proinflammatory cytokines via signaling pathways involving STAT1 and IFN regulatory factor-1

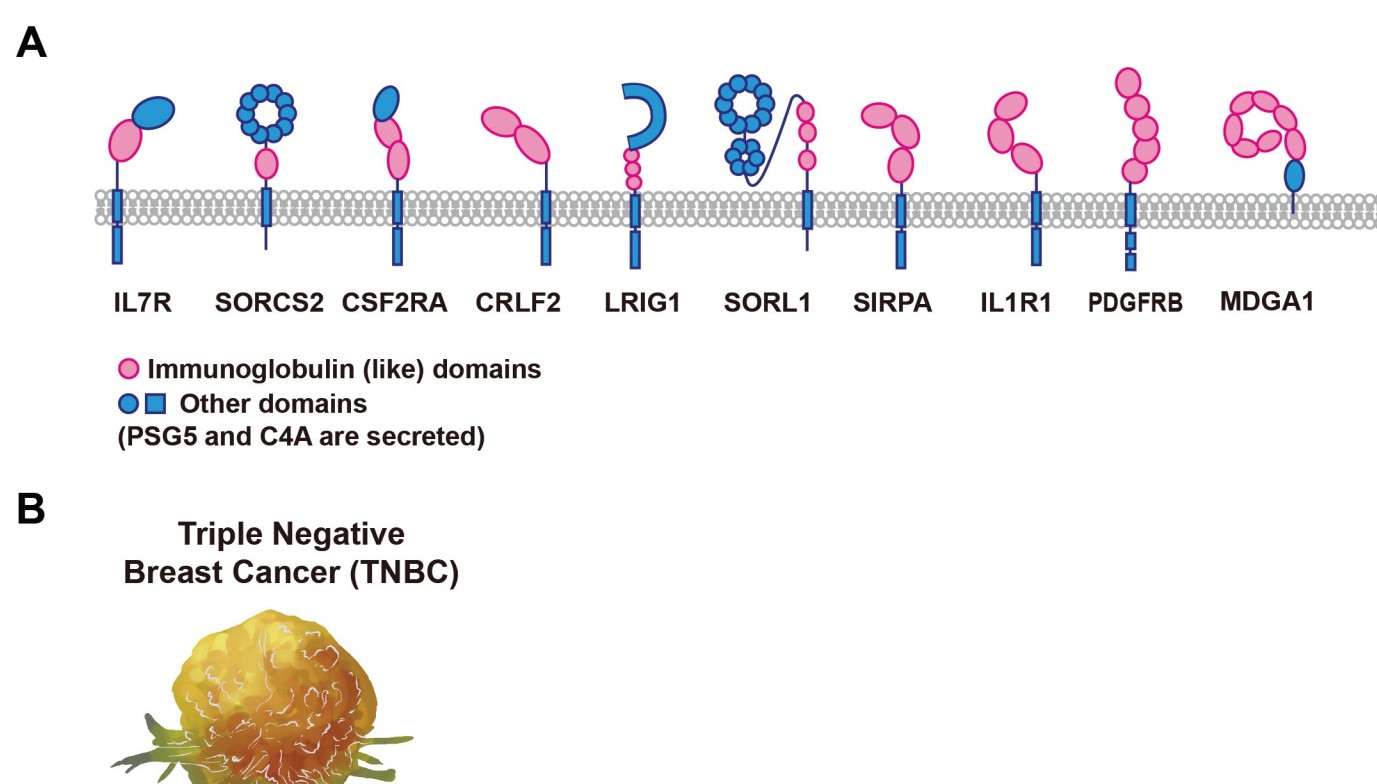

**A**

IL7R  SORCS2  CSF2RA  CRLF2  LRIG1  SORL1  SIRPA  IL1R1  PDGFRB  MDGA1

● Immunoglobulin (like) domains
● ■ Other domains
(PSG5 and C4A are secreted)

**B**

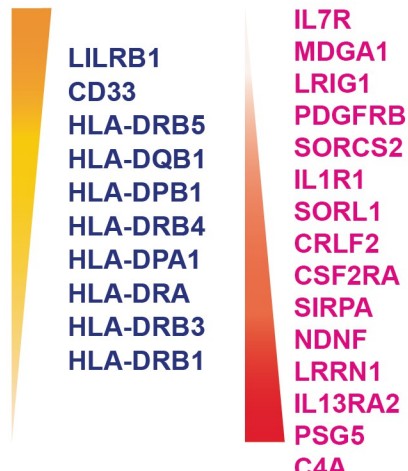

**Triple Negative
Breast Cancer (TNBC)**

**Downregulation**   **Upregulation**

| Downregulation | Upregulation |
|---|---|
| LILRB1 | IL7R |
| CD33 | MDGA1 |
| HLA-DRB5 | LRIG1 |
| HLA-DQB1 | PDGFRB |
| HLA-DPB1 | SORCS2 |
| HLA-DRB4 | IL1R1 |
| HLA-DPA1 | SORL1 |
| HLA-DRA | CRLF2 |
| HLA-DRB3 | CSF2RA |
| HLA-DRB1 | SIRPA |
| | NDNF |
| | LRRN1 |
| | IL13RA2 |
| | PSG5 |
| | C4A |

**Fig 5.** A, Schematic illustration depicting the structure of extracellular domains of genes analyzed as upregulated in RNA sequencing. B, List of genes with confirmed increases or decreases between Ori and TW1, validated through RNA sequencing and qRT-PCR.

(IRF-1) [59]. PD-1/PD-L1 blockade currently stands as one of the most prominent immuno-therapeutic approaches. Contemporary strategies within the PD-1/PD-L1 pathway aspire to predict and stratify patients receiving ICIs by utilizing checkpoint interaction status as a mechanism-based biomarker, with the ultimate objective of inhibiting these interactions. Recent

investigations have unveiled the occurrence of posttranslational modifications (PTMs), such as ubiquitination, glycosylation, phosphorylation, palmitoylation, and acetylation, in PD-L1 and PD-1. These PTMs wield significant influence over PD-L1 stability, PD-1/PD-L1 interactions, and the cellular translocation of PD-L1, underlining the critical importance of the PD-1/PD-L1 colocation score in predicting responses to anti-PD-1/PD-L1 immunotherapy.

While structural similarities exist among immunoglobulins, discrepancies in PTMs can introduce variations in binding partners or functional outcomes. Beyond the well-established immune checkpoint regulators exemplified by PD-1/PD-L1, this research aims to uncover new biomarkers, including novel immunoglobulins, capable of interacting with immune checkpoint regulators on immune cells. In this study, we have identified changes in various Ig genes through cancer deformation. We aspire that our approach will facilitate the discovery of new immunotherapy biomarkers.

## Acknowledgments

We thank the patients and their families. We also wish to convey our profound appreciation to Dr. Paul Nghiem and his lab members for helpful discussions.

## Author Contributions

**Conceptualization:** Se Min Kim, JuKyung Lee, Jung Hyun Lee.

**Data curation:** Se Min Kim, Namu Park, Jung Hyun Lee.

**Formal analysis:** Hye Bin Park, Changho Chun, Kyung Hoon Kim, Hyung Jin Kim, Jung Hyun Lee.

**Funding acquisition:** Jong Seob Choi, Jung Hyun Lee.

**Investigation:** Se Min Kim, Namu Park, Jung Hyun Lee.

**Methodology:** Se Min Kim, Jung Hyun Lee.

**Project administration:** Se Min Kim, Jung Hyun Lee.

**Resources:** Jung Hyun Lee.

**Supervision:** Jung Hyun Lee.

**Validation:** Se Min Kim, Jung Hyun Lee.

**Visualization:** Se Min Kim, Jung Hyun Lee.

**Writing – original draft:** Se Min Kim, Jung Hyun Lee.

**Writing – review & editing:** Se Min Kim, Namu Park, Hye Bin Park, JuKyung Lee, Changho Chun, Kyung Hoon Kim, Jong Seob Choi, Hyung Jin Kim, Sekyu Choi, Jung Hyun Lee.

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
