## [Decision Letter · Decision Letter 0]

28 Dec 2023

PONE-D-23-37253Exploring New Biomarker Candidates for Immunotherapy through Cancer Deformation-Induced Gene ChangesPLOS ONE

Dear Dr. Lee,

Thank you for submitting your manuscript to PLOS ONE. After careful consideration, we feel that it has merit but does not fully meet PLOS ONE’s publication criteria as it currently stands. Therefore, we invite you to submit a revised version of the manuscript that addresses the points raised during the review process.

We look forward to receiving your revised manuscript.

Kind regards,

Shigao Huang

Academic Editor

PLOS ONE

 [the Elsa U Pardee Foundation (to J.H.L), MCC patient gift fund (to J.H.L)].  

[This work was supported by the Elsa U Pardee Foundation (to J.H.L), MCC patient gift fund (to J.H.L), the Mirae Asset Park Hyeon Joo Foundation (to S.M.K). ]

 [the Elsa U Pardee Foundation (to J.H.L), MCC patient gift fund (to J.H.L)]. 

6. Please provide a complete Data Availability Statement in the submission form, ensuring you include all necessary access information or a reason for why you are unable to make your data freely accessible. If your research concerns only data provided within your submission, please write "All data are in the manuscript and/or supporting information files" as your Data Availability Statement.

7. We note that Figure(s) 1A, 2B and 5 in your submission contain copyrighted images. All PLOS content is published under the Creative Commons Attribution License (CC BY 4.0), which means that the manuscript, images, and Supporting Information files will be freely available online, and any third party is permitted to access, download, copy, distribute, and use these materials in any way, even commercially, with proper attribution. For more information, see our copyright guidelines: http://journals.plos.org/plosone/s/licenses-and-copyright.

a. You may seek permission from the original copyright holder of Figure(s) 1A, 2B and 5 to publish the content specifically under the CC BY 4.0 license. 

8. Please remove your figures from within your manuscript file, leaving only the individual TIFF/EPS image files, uploaded separately. These will be automatically included in the reviewers’ PDF.

Reviewers' comments:

Reviewer's Responses to Questions

**Comments to the Author**

1. Is the manuscript technically sound, and do the data support the conclusions?

Reviewer #1: Partly

Reviewer #2: Partly

2. Has the statistical analysis been performed appropriately and rigorously? 

Reviewer #1: Yes

Reviewer #2: Yes

3. Have the authors made all data underlying the findings in their manuscript fully available?

Reviewer #1: Yes

Reviewer #2: No

4. Is the manuscript presented in an intelligible fashion and written in standard English?

Reviewer #1: No

Reviewer #2: Yes

5. Review Comments to the Author

Reviewer #1: In the present paper, authors investigated genes expressing Ig-like domains as potential biomarkers of immunotherapy in TNBC. Manuscript is potentially interesting however, an extensive english revision is needed for a better comprehension of the presented results.

Herein a few comments:

1) Results section should be more concise and should include only the authors findings; the literature data should be included in a separate section.

2) A brief paragraph on breast cancer subtype (luminal A, B, Her2 TNBC) should be included in the introduction.

3) PD-L1 pathway in TNBC as well as most relevant immunotherapy trials in brest cancers should be discussed.

Reviewer #2: Dear,

Thank you for submitting the manuscript entitled "Exploring New Biomarker Candidates for Immunotherapy through Cancer Deformation-Induced Gene Changes". I was very interested in your data on MHC (HLA).

I think these findings should be published for the future research, but also have some concerns below. Hope to see you soon.

1. You present the data of only Ori and TW1 at figure 3A, 4A and 4B, but it seems you have 3 results each. Did you combine Ori1-2-3? Where did TW2 and TW3 go? Why did you focus on TW1?

Same matter on figure 3B and 4B. Why you have only TW1 and TW2? What is the differences between TW1-2-3?

2. If your "previous research" on page 4 and 5 were #50 on page6, please insert the reference.

3. Font of some part is different (smaller) from the rest. (page 6 and 10)

Sincerely,

6. PLOS authors have the option to publish the peer review history of their article (what does this mean?). If published, this will include your full peer review and any attached files.

Reviewer #1: No

Reviewer #2: **Yes: **Akira I. Hida

---

## [Author Response · Author response to Decision Letter 0]

17 Jan 2024

Response to Reviewers

We sincerely appreciate your meticulous review of our manuscript. Your feedback is precious to us, and we have carefully incorporated your insightful suggestions into the revised version. Your thorough observations have significantly improved the quality and clarity of our research. We greatly respect the time and effort you have devoted to evaluating our work, and we eagerly anticipate receiving your esteemed feedback on the revised manuscript.

Reviewer #1: 

In the present paper, authors investigated genes expressing Ig-like domains as potential biomarkers of immunotherapy in TNBC. Manuscript is potentially interesting however, an extensive english revision is needed for a better comprehension of the presented results.

Herein a few comments:

1) Results section should be more concise and should include only the authors findings; the literature data should be included in a separate section.

Response: We appreciate your valuable feedback. We have revised and improved the Results section of this manuscript accordingly. You can review the updated content in the manuscript tracking.

2) A brief paragraph on breast cancer subtype (luminal A, B, Her2 TNBC) should be included in the introduction.

Response: Thank you for your valuable comments. As per your advice, we have added content regarding the breast cancer subtype to the introduction.

3) PD-L1 pathway in TNBC as well as most relevant immunotherapy trials in breast cancers should be discussed.

Response: We are grateful for your invaluable feedback. We have incorporated the significance of PD-1/PD-L1 pathway-based treatment in triple-negative breast cancer (TNBC) into the introductory section of our results.

Reviewer #2:

Thank you for submitting the manuscript entitled "Exploring New Biomarker Candidates for Immunotherapy through Cancer Deformation-Induced Gene Changes". I was very interested in your data on MHC (HLA).

I think these findings should be published for the future research, but also have some concerns below. Hope to see you soon.

1. You present the data of only Ori and TW1 at figure 3A, 4A and 4B, but it seems you have 3 results each. Did you combine Ori1-2-3? Where did TW2 and TW3 go? Why did you focus on TW1?

Same matter on figure 3B and 4B. Why you have only TW1 and TW2? What is the differences between TW1-2-3?

Response: Thank you for your valuable feedback, and we apologize for any confusion regarding the sample names. The cells we designated as 'Ori' are MDA-MB 231 cells. Cells that underwent cancer deformation after passing through a 3μm pore size were named 'TW1,' and those that underwent further cancer deformation after TW1 were named 'TW2.' RNA sequencing experiments were conducted using Ori and TW1 cells. However, qPCR validation experiments were performed on Ori, TW-1, and TW-2. We have now made revisions in the results section to make this more transparent and less confusing.

2. If your "previous research" on page 4 and 5 were #50 on page6, please insert the reference.

Response: We sincerely appreciate your valuable feedback. Your points were accurate, and we have cited the relevant literature appropriately in the revised manuscript for your review.

3. Font of some part is different (smaller) from the rest. (page 6 and 10)

Response: Thank you for thoroughly reviewing our manuscript. We have gone through the entire formatting once again.

---

## [Decision Letter · Decision Letter 1]

25 Apr 2024

Exploring Novel Immunotherapy Biomarker Candidates Induced by Cancer Deformation

PONE-D-23-37253R1

Dear Dr. Hyun Lee

We’re pleased to inform you that your manuscript has been judged scientifically suitable for publication and will be formally accepted for publication once it meets all outstanding technical requirements.

Kind regards,

Afsheen Raza, PhD

Academic Editor

PLOS ONE

Additional Editor Comments (optional):

Reviewers' comments:

Reviewer's Responses to Questions

**Comments to the Author**

1. If the authors have adequately addressed your comments raised in a previous round of review and you feel that this manuscript is now acceptable for publication, you may indicate that here to bypass the “Comments to the Author” section, enter your conflict of interest statement in the “Confidential to Editor” section, and submit your "Accept" recommendation.

Reviewer #2: All comments have been addressed

2. Is the manuscript technically sound, and do the data support the conclusions?

Reviewer #2: Yes

3. Has the statistical analysis been performed appropriately and rigorously? 

Reviewer #2: Yes

4. Have the authors made all data underlying the findings in their manuscript fully available?

Reviewer #2: Yes

5. Is the manuscript presented in an intelligible fashion and written in standard English?

Reviewer #2: Yes

6. Review Comments to the Author

Reviewer #2: (No Response)

7. PLOS authors have the option to publish the peer review history of their article (what does this mean?). If published, this will include your full peer review and any attached files.

Reviewer #2: No

---

## [Editor Report · Acceptance letter]

2 May 2024

PONE-D-23-37253R1 

PLOS ONE

Dear Dr. Lee, 

I'm pleased to inform you that your manuscript has been deemed suitable for publication in PLOS ONE. Congratulations! Your manuscript is now being handed over to our production team.

Kind regards, 

on behalf of

Dr. Afsheen Raza 

Academic Editor

PLOS ONE